Review article

# Warming of $+1.5\,°C$ is too high for polar ice sheets
Chris R. Stokes [1] ✉, Jonathan L. Bamber [2], Andrea Dutton [3] & Robert M. DeConto [4]

Mass loss from ice sheets in Greenland and Antarctica has quadrupled since the 1990s and now represents the dominant source of global mean sea-level rise from the cryosphere. This has raised concerns about their future stability and focussed attention on the global mean temperature thresholds that might trigger more rapid retreat or even collapse, with renewed calls to meet the more ambitious target of the Paris Climate Agreement and limit warming to $+1.5\,°C$ above pre-industrial. Here we synthesise multiple lines of evidence to show that $+1.5\,°C$ is too high and that even current climate forcing ($+1.2\,°C$), if sustained, is likely to generate several metres of sea-level rise over the coming centuries, causing extensive loss and damage to coastal populations and challenging the implementation of adaptation measures. To avoid this requires a global mean temperature that is cooler than present and which we hypothesise to be closer to $+1\,°C$ above pre-industrial, possibly even lower, but further work is urgently required to more precisely determine a 'safe limit' for ice sheets.

Global mean sea level (GMSL) increased by around 20 cm from 1901 to 2018, with the rate of change accelerating from ~1.4 mm year$^{-1}$ (1901–1990) to ~3.7 mm year$^{-1}$ (2006–2018)[1] and, most recently, to 4.5 mm year$^{-1}$ (2023)[2]. Several processes contribute to sea-level rise (SLR), but melting of glaciers and ice sheets is the dominant source, adding ~1.6 mm year$^{-1}$ from 2006 to 2018, and now exceeding thermal expansion of the oceans (~1.4 mm year$^{-1}$)[1]. Smaller mountain glaciers and ice caps dominated the cryosphere's contribution to GMSL rise during the 20th Century[1,3–5] and they will continue to shrink rapidly[1,6], with profound impacts on water resources and human activities. However, their cumulative volume in terms of sea-level equivalent (SLE: ~32 cm)[7] is dwarfed by the Earth's ice sheets in Greenland (GrIS: 7.4 m SLE)[8], West Antarctica (WAIS: 5.3 m SLE)[9] and East Antarctica (EAIS: 52.2 m SLE)[9]. Of major concern is that the combined sea-level contribution from ice sheets (~11.9 mm from 2006 to 2018) is now larger than for mountain glaciers and ice caps (~7.5 mm over the same period) and is four times higher than in the 1990s[1]. Furthermore, these trends are set to continue, with the most recent IPCC projections[1] estimating a combined ice sheet contribution at 2100 ranging from $+4$ to $+37$ cm under a low emissions scenario (Shared Socioeconomic Pathway: SSP1-2.6) to $+12$ to $+52$ cm under the high SSP5-8.5 scenario; but they could not rule out low confidence projections of total sea level rise under SSP5-8.5 that might exceed $+15$ m at 2300. Hence, continued mass loss from ice sheets poses an existential threat to the world's coastal populations, with an estimated[10] one billion people inhabiting land less than 10 m above sea level and around 230 million living within 1 m.

Without adaptation, conservative estimates[11] suggest that 20 cm of SLR by 2050 would lead to average global flood losses of US$1 trillion or more per year for the world's 136 largest coastal cities. Furthermore, whilst absolute values of SLR are useful, it is the rate of change that is likely to determine the appropriate societal response, with the IPCC[12] suggesting that 'very high rates' of SLR (e.g. 10–20 mm year$^{-1}$) would challenge the implementation of adaptation measures that involve long lead times. Such high rates (e.g. >10 mm year$^{-1}$) might occur as early as 2100 if the recent acceleration in SLR continues throughout this century[2].

The 'threat of disaster' posed by the world's ice sheets has been recognised for some time[13] but recent observations of mass loss[14–19], coupled with advances in numerical ice sheet modelling, have seen an increased emphasis on projecting their future contribution to SLR[20–30], with a particular focus on the different SSPs used to force climate models and specific temperature targets, especially the Paris Climate Agreement, which aims to limit warming to well below $+2\,°C$ above the pre-industrial baseline (1850–1900) and, ideally, $+1.5\,°C$. Note that the IPCC refer to these globally-averaged temperature thresholds in terms of a 20-year mean relative to pre-industrial, such that briefly exceeding $+1.5\,°C$, for example (as observed in 2024, which became the first year with an average surface air temperature exceeding $+1.5\,°C$)[31], does not constitute breaching that limit. Instead, the year of exceedance is defined as the mid-point in the 20-year period with an average global temperature above $+1.5\,°C$. A limitation of this approach is that exceedance is only recognised a decade after crossing the threshold, risking a delay in both recognising and responding to such an event[32].

[1]Department of Geography, Durham University, Durham, UK. [2]School of Geographical Science, University of Bristol, Bristol, UK. [3]Department of Geoscience, University of Wisconsin–Madison, Madison, WI, USA. [4]Department of Earth, Geographic, and Climate Sciences, University of Massachusetts Amherst, Amherst, MA, USA. ✉e-mail: c.r.stokes@durham.ac.uk

## Box 1 | self-reinforcing feedback mechanisms leading to rapid ice sheet retreat

In a warming climate, continental ice sheets are vulnerable to self-reinforcing feedback mechanisms that generate rapid ice loss and rates of SLR that could be around ten times higher than present (e.g. 40 mm year[−1]). The first of these is the (i) 'surface elevation and melt feedback', whereby an initial lowering of the ice sheet surface exposes it to warmer air at lower elevations, increasing melt rates, lowering the surface more rapidly, and further increasing melting. This feedback is thought to have caused the rapid 'collapse' of part of the North American Ice Sheet complex during the last deglaciation, contributing almost 4 m per century to GMSL[41,42], and some suggest that central-west Greenland may already be close to a critical transition under current climate forcing[50]. The second key feedback mechanism is known as (ii) 'marine ice sheet instability' (MISI), which may occur when the ice sheet rests on bedrock below sea level that deepens towards its interior (a retrograde slope). Initial retreat (e.g. triggered by warm ocean currents thinning the floating portions of the ice sheet near the grounding line) leads to an increase in ice thickness and hence ice discharge at the grounding line. This forces the grounding line into deeper water with thicker ice, increasing the cross-sectional area at the grounding line and increasing ice discharge into the ocean[149]. Scientists warned that the WAIS was particularly vulnerable to MISI in the 1970s[13] and some suggest it may already be underway[52,53], but large parts of the EAIS also sit on retrograde slopes and are similarly susceptible[29,100,101]. The third potential feedback is (iii) 'marine ice cliff instability' (MICI), which has been hypothesised more recently[21,27,150]. This mechanism is initiated when buttressing ice shelves are rapidly removed (e.g. through hydrofracturing driven by surface melting)[150], exposing large ice cliffs at the grounding line that are mechanically unstable[151] and progressively collapse due to increasing ice thicknesses[150]. The validity and implementation of MICI is debated[1,152] and, unlike MISI, there is no evidence that it is currently underway, with one recent study showing it might not occur this century[152]. However, the inclusion of this process in ice sheet modelling has helped reconcile Antarctica's contribution to high sea-levels during past warm periods[21,150] and, although the IPCC assess future projections with MICI as *low confidence*, they were not ruled out[1].

A key point in relation to each feedback mechanism is that a relatively small perturbation in climate (warming) can initiate a large and rapid response from the ice sheet that is, essentially, irreversible on human timescales (i.e. collapse evolves over centennial time-scales but regrowth would take several millennia). Furthermore, there is likely to be a lag in response to the applied forcing, such that: (a) the climate warming that has already taken place (and will undoubtedly continue in the short-term) may have already committed us to future instability; and (b) we may not know that a temperature threshold has triggered an instability until it is well underway.

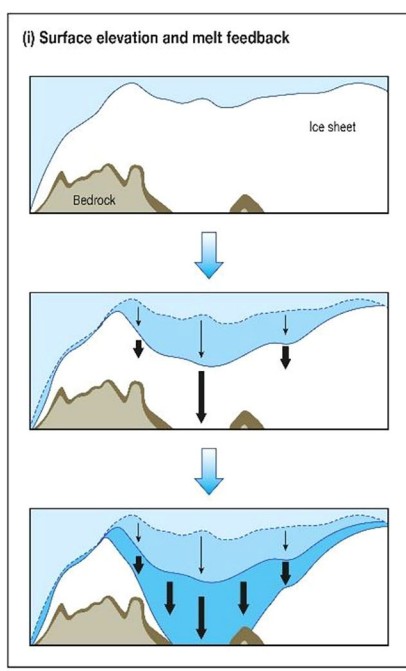

(i) Surface elevation and melt feedback

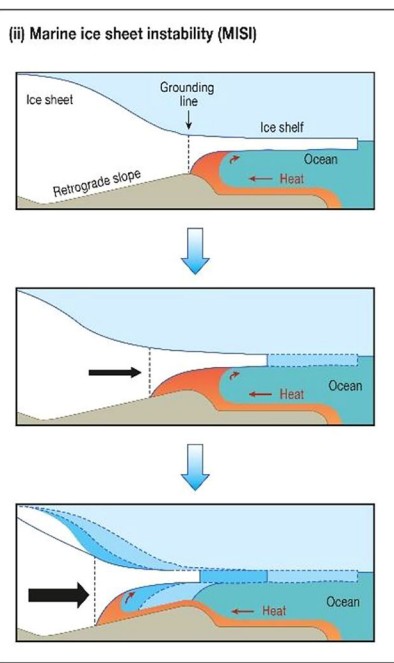

(ii) Marine ice sheet instability (MISI)

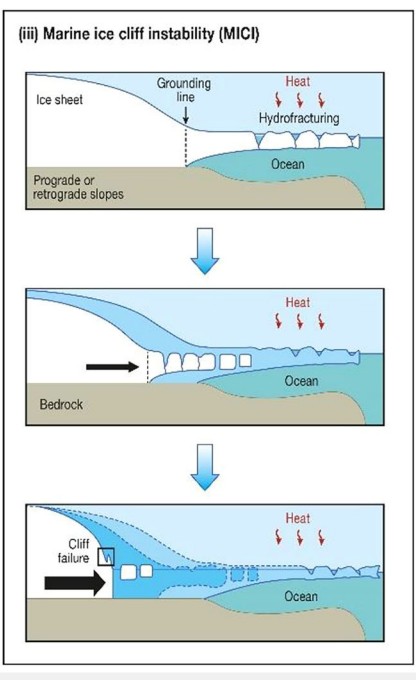

(iii) Marine ice cliff instability (MICI)

Future projections of ice sheet change using numerical modelling are informed, often qualitatively but sometimes quantitatively, by palaeo-records of ice sheet configuration and sea level from periods when Earth's climate conditions were similar or slightly warmer than present[21,33–37]. During the Last Interglacial (LIG: ~129–116 ka), for example, global mean surface temperatures were +0.5 to +1.5 °C higher than pre-industrial, $CO_2$ concentrations were 266–282 ppm, but GMSL was likely several metres higher than present[1,38]. Palaeo-records and numerical modelling also demonstrate that ice sheets exhibit highly non-linear responses to climate forcing and are sensitive to temperature thresholds that, once crossed, lead to rates of SLR much higher than present[24,27,34,39,40]. During the last deglaciation, for example, GMSL rise during Meltwater Pulse 1 A (around 14.7 ka) peaked at 3.5–4 m per century[39,41,42], i.e. around ten times higher than present-day, albeit with a larger number of palaeo-ice sheets than at present. Such rapid rates of SLR (centimetres per year) are thought to result from self-reinforcing feedbacks (Box 1) that may be triggered by only a small increase in temperature but which propel the system into a new and contrasting state which requires a much larger decrease in temperature to return to the original state[24]. Furthermore, it takes much longer for ice sheets to regrow (tens of thousands of years) than to retreat (centuries to millennia).

**Table 1 | Climate conditions and ice sheet contributions to global mean sea level during three past warm periods that are often used as analogues for current or near-future climate conditions: Marine Isotope Stage (MIS) 5e, MIS 11 and the Mid Pliocene Warm Period (MPWP). Published data sources used to compile the Table are available in Supplementary Table 1**

| | | Marine Isotope Stage 5e (~129–116 ka) | Marine Isotope Stage 11 (~400 ka) | Mid-Pliocene Warm Period (3.3–2.9 Ma) |
|---|---|---|---|---|
| Forcing (peak values) | $CO_2$ | 287 ppm | 286 ppm | 350–450 ppm |
| | Global mean temp. anomaly[b] | 1 °C | 1–2? °C | 2–5 °C |
| | 'Arctic' temp. anomaly[c] | 4–8 °C | 4–10 °C | 4–11 °C |
| | 'Antarctic' temp. anomaly[c] | 4–5 °C | 2–3 °C | ? °C |
| Greenland | Ice sheet response | Central portion of ice sheet largely intact; southern GrIS deglaciated | NW and southern GrIS deglaciated; complete deglaciation possible | Partial to complete deglaciation |
| | GMSL contribution | Estimates range from 1 to >5 m | Estimates include: >1.4 m, 4.5–6 m; 6.1 (3.9–7) m | Up to 7.4 m |
| Antarctica | Ice sheet response | WAIS retreat, some EAIS retreat possible | EAIS retreat & increased runoff indicated | Retreat in WAIS and EAIS |
| | GMSL contribution | Debated, up to 5 m (or higher if EAIS contributed) | Uncertain; up to 3–4 m | Possibly 13–17 m |
| Peak GMSL | | 2–9 m above present | 6–13 m above present | 10–20 m above present |
| Interpretation[a] | | GMSL exceeded present (H); A portion of the GrIS retreated (H); AIS contributed to GMSL rise (M) | GMSL exceeded present (M-H); GrIS smaller than present (H); majority of the GrIS melted (M-H); AIS smaller than present (M) | GrIS smaller than present (H); AIS smaller than present (H) |

[a]Low (L), Medium (M) and High (H) confidence based on a qualitative literature survey (not statistical analysis).
[b]Rounded to the nearest whole number relative to pre-industrial baseline (1850–1900).
[c]Temperature anomalies relative to pre-industrial are derived from point data; localities of data may vary between time intervals so may not be directly comparable.

A key implication of this hysteresis behaviour[24] is that current rates of SLR could increase rapidly with only small changes in temperature. Identifying temperature thresholds for each ice sheet is, therefore, critically important, with recent work[29,43,44], suggesting best estimates for the GrIS and WAIS at +1.5 °C and marine-based sectors of the EAIS at +2 to +3 °C above pre-industrial. This is consistent with the IPCC's assessment[45] that ice sheet instabilities could be triggered somewhere between +1.5 and +2 °C, leading to renewed calls for policymakers to target the lower limit of the Paris Climate Agreement and 'keep +1.5 °C alive'[46,47]. Some of the most serious potential consequences of exceeding this value (e.g. complete loss of the GrIS) can be avoided if global mean temperatures are quickly reduced to below +1.5 °C[40]. However, it has also been noted that even temporarily overshooting such thresholds could lead to several metres of SLR[27,40,48,49], with one study[22] finding that median SLR in 2300 is 4 cm higher per decade of overshoot above +1.5 °C, even under a range of net-zero scenarios. In contrast, others have argued that such temperature thresholds may be lower than +1.5 °C and may already be close to exceedance for parts of Greenland[50] and the WAIS[51], where some argue that marine ice sheet instability (Box 1) is potentially already underway[52,53].

Here, we synthesise evidence from the past warm periods, recent observations of ice sheet mass balance and numerical modelling to show that +1.5 °C is far too high and that even current climate conditions (+1.2 °C above pre-industrial), if sustained, could trigger rapid ice sheet retreat and high rates of SLR (e.g. >10 mm year$^{-1}$) that would stretch the limits of adaptation. Given the catastrophic consequences for humanity of a rapid collapse of one or more ice sheets leading to multi-metre SLR, we conclude that adopting the precautionary principle is imperative and that a global mean temperature cooler than present is required to keep ice sheets broadly in equilibrium. Precisely determining a 'safe limit' for ice sheets is challenging, not least because so few studies have projected their response to cooler-than-present climate conditions but based on this review, we hypothesise that it probably lies close to, or even below, +1.0 °C above pre-industrial.

## Ice sheet contributions to sea level during past warm periods

Palaeo-records provide important empirical benchmarks to understand how ice sheets respond to various warming scenarios and help calibrate ice sheet models used to project future SLR[21,27,33–36]. Geologically recent warm periods such as the LIG (Marine Isotope Stage 5e: ~125 ka), MIS 11 (~400 ka) and the mid-Pliocene warm period (MPWP: 3.3–2.9 Ma) are often targeted because global mean temperatures were similar to, or slightly warmer than, present; and with atmospheric $CO_2$ concentrations ranging from lower than, to similar to, present (~280–450 ppm) (Table 1).

### Last interglacial

There is broad consensus that GMSL during the LIG was higher than present (Fig. 1a), and emerging evidence that peak contributions from Greenland and Antarctica may have been out of phase[54–56]. An earlier assessment of peak GMSL placed the most likely estimate at +6 to +9 m above present[38]. Subsequently, two studies from the Bahamas[57,58] placed this estimate considerably lower (+1 to +4 m: Fig. 1a), while another using coastal deposits from northern Europe[56] requires around +6 m SLE (+ 3.6 to +8.7 m) from Antarctica alone. Additional uncertainties relating to GMSL rise during the LIG stem from the recognition that dynamic topography resulting from mantle convection can induce metre-scale changes in land surface elevation that may bias estimates of past sea level[59]. Accounting for lateral variability in both viscosity and lithospheric thickness also modifies interpretations of GMSL from site-specific data[60], but due to the computational intensity of 3D Glacial-Isostatic Adjustment (GIA) modelling, most studies use 1D GIA models. Thus, uncertainties associated with the magnitude of peak LIG GMSL have increased over the last decade.

Despite uncertainties, there is high confidence that the GrIS partially retreated during the LIG, based on evidence from ice cores, deep-sea sediment cores, and ice sheet models (Table 1). Estimates of the magnitude of

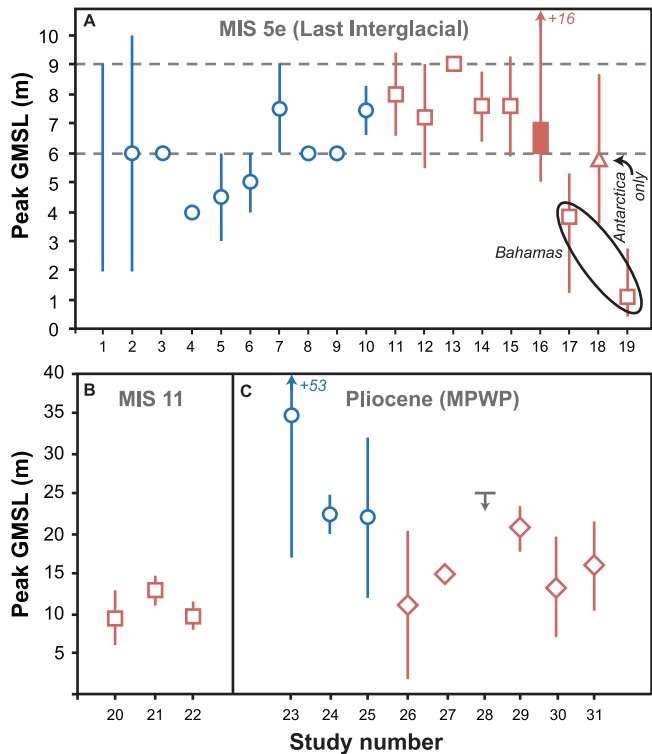

**Fig. 1 | Published estimates of peak global mean sea level (GMSL) from coastal sedimentary archives during past warm periods. A** the Last Interglacial (MIS 5e); **B** MIS 11; and **C** the Mid-Pliocene Warm Period. Estimates are plotted from left to right in chronological order of publication, numbered along the x-axis, with corresponding references listed in Supplementary Table 2). Symbols represent GMSL estimates with no GIA correction (blue circles); data combined with GIA modelling (red squares); GIA and dynamic topography modelling (red diamonds); Antarctic only contribution with GIA correction (red triangle, labelled with arrow); a preferred estimate of GIA corrected output (solid red rectangle); and a maximum estimate based on an analysis of amplitude of sea-level change from marine sediments (grey bar with downward pointing arrow). Horizontal dashed lines in (**A**) denote the +6 to +9 m range proposed in a previous assessment[38].

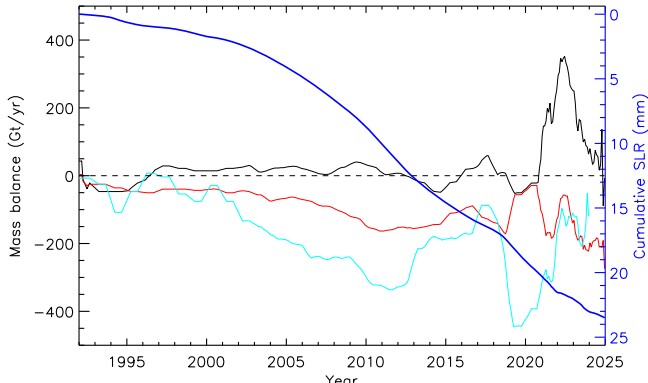

**Fig. 2 | Recent ice sheet mass balance estimates from 1992–2024.** Data from 1992–2020 are taken from the latest Ice Sheet Mass Balance Inter-comparison Exercise[18], updated with satellite gravimetry data[148] from GRACE for 2021–2024 inclusive, with East Antarctic Ice Sheet (EAIS) in black, West Antarctic Ice Sheet (WAIS) in red and Greenland Ice Sheet (GrIS) in turquoise (to end of 2023). The cumulative sea-level rise equivalent from the GrIS and WAIS is shown in the thicker blue line.

retreat generally vary between +2 to +5 m SLE. While there less agreement about whether a part (or which part) of the AIS retreated, an increasing number of studies argue that Antarctica contributed substantively to LIG SLR, including evidence from ice cores[61], deep-sea sediment records[55]; coral reefs[54], coastal sediments[56], and even octopus DNA[62]. Notably, ice sheets were responding to LIG temperatures that were either similar to present, or within the range expected in the next few decades for polar regions[61,63,64], implying that +1 to +1.5 °C world would generate several metres of SLR (Fig. 1a; Table 1).

There are additional insights from the LIG relevant to understanding ice sheet response to present and future warming. First, if the AIS and GrIS retreated (and regrew) out of phase, as has been hypothesised[54–56], this would have buffered the total amount of SLR experienced at any one time. Hence, relying on the peak in LIG GMSL as an analogue for future warming may underestimate ice sheet retreat that is currently observed simultaneously in both hemispheres due to increasing greenhouse gas concentrations, rather than orbital forcing. Second, while there has been much focus on determining peak GMSL and peak temperatures for the LIG, this may obscure more important thresholds for ice sheets that trigger retreat well before peak climate forcing is experienced. Third, sedimentary evidence for multiple peaks in LIG sea level imply a dynamic cryosphere capable of producing meter-scale, stepwise pulses in SLR on centennial timescales[65,66]. Such high rates of ice sheet retreat (centimetres of SLR per year) may be beyond the limits of adaptation and have profound implications for coastal populations[12].

## MIS 11 and the mid-Pliocene warm period (MPWP)

Earlier warm periods are also instructive for understanding ice sheet response to prolonged but relatively weak climate forcing (e.g. MIS 11) and to atmospheric $CO_2$ levels that are comparable to present (e.g. the MPWP). During MIS 11, there is evidence for large-scale deglaciation in northwest[37] and southern Greenland[67]. The chemistry of subglacial precipitates also suggests that part of the EAIS retreated by +3 to +4 m SLE[68]. These findings are compatible with GMSL estimates ranging from +6 to +13 m above present during MIS 11 (Fig. 1b)[69,70]. Regional summer insolation anomalies during MIS 11 were weaker than during the LIG and higher sea-levels have, therefore, been attributed to its much longer duration[71]. This is important in the context of future ice sheet response, i.e. even if global mean temperature can be stabilised at +1.5 °C in the long-term (perhaps after a brief overshoot), it would expose ice sheets to prolonged warmth, rather than the more transient warming that characterised most interglacials driven by varying insolation from orbital forcing. Hence, it has been argued[71] that using interglacial temperatures alone may underestimate the ice sheet response to a more prolonged exposure to the same climate state.

The most recent interval when atmospheric $CO_2$ levels were similar to present was the MPWP. Peak GMSL remains uncertain[38,72–75], but most estimates fall within +10 to +20 m (Fig. 1c, Table 1), implying an ice-free Greenland and an Antarctic contribution ranging from +3 to +13 m SLE (or more), likely sourced from the WAIS but with higher values implicating marine-based parts of the EAIS[29]. Notwithstanding the inherent uncertainties of palaeo-records, they clearly indicate that if ice sheets are required to equilibrate to a world with global mean temperatures between +1.0 and +1.5 °C, and $CO_2$ concentrations between 350 and 450 ppm, policymakers should prepare for several metres of GMSL-rise over centennial to millennial time-scales[76].

## Recent observations of ice sheet mass balance

The advent of satellite remote sensing in the 1970s enabled some of the first estimates of ice sheet mass balance[77], with systematic observations commencing in 1992[78–81], followed by a major expansion in the diversity of measurement techniques and an emphasis on reconciling data from different sensors[15,18,82]. Despite marked interannual variability[83], these records (Fig. 2) show that all three ice sheets were close to balance in the early 1990s[15,18,78–83]. Since the mid-1990s, however, there is a clear trend of mass loss from both the GrIS and WAIS when averaged over the last three decades (Fig. 2); whereas the EAIS has remained close to balance, albeit with much larger uncertainties and with a larger spread of estimates from different methods[18,29]. The combined effect is that ice sheets increased GMSL by

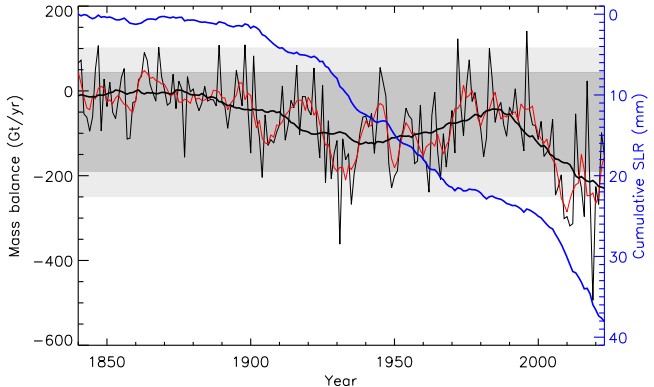

**Fig. 3 | Reconstruction of the GrIS mass balance (1840-2012) from re-analyses and historical observations[92], combined with GRACE data[148] for 2013–2023 inclusive.** The annual mass balance time series is shown as the thin black line, with a 5-year (red) and 30-year (thick black line) running mean. The cumulative sea-level rise (SLR) is shown in blue. The dark grey and light grey shading show the 2 and 3 sigma standard deviations, respectively, of the 5-year running means with the 182-year mean removed.

+21 ± 1.9 mm between 1992 and 2020, with the rate of loss increasing from 105 Gt year⁻¹ (1992–1996) to 372 Gt year⁻¹ (2016–2020)[18], and meaning they are now the dominant contributor to barystatic SLR[1]. Indeed, it has been noted[84] that mass loss from 2007 to 2017 tracked the upper range projected in the IPCC's Fifth Assessment Report (AR5)[85]; and that if high rates of loss were to continue (+1.2 ± 0.2 mm year⁻¹ between 2019 and 2020), they would track above the upper range projected in AR6 (+1.0 to +1.1 mm year⁻¹ for the current decade)[18]. Furthermore, by neglecting the retreat of marine-terminating outlet glaciers, assessments of mass balance[1,18] may have underestimated mass loss from Greenland by as much as 20%[19].

The long-term trend of increasing mass loss from the GrIS and WAIS (Fig. 2) raises two important questions: (i) is there a statistically significant acceleration; and (ii) to what extent is mass loss due to external (anthropogenic) forcing versus internal ('natural') variability? Answering the former is difficult because satellite observations are likely too short to separate a long-term acceleration from short-term ice sheet variability[86], although some have inferred statistically significant accelerations in both Greenland and the WAIS[14,87]. Irrespective, mass loss from the GrIS is particularly striking in a long-term context (Fig. 3) and it dominates the ice-sheet contributions to GMSL rise, with a notable increase in surface melting[17,50,88] that is expected to become increasingly important in the future[25,28]. This is not surprising given that the Arctic has warmed nearly four times faster than the global average since 1979[89], a phenomenon known as Arctic Amplification. Most Coupled Model Intercomparison Project (CMIP) Earth System Models do not capture this rate of Arctic Amplification[89], meaning that numerical ice sheet models may underestimate the future response of the GrIS to changes in global mean temperature[90].

Answering the second question is also challenging due to the short observational record and because ice sheets respond over timescales from diurnal (locally) to millennial[83], making it hard to detect naturally occurring low-frequency variability[91]. Nevertheless, reconstructions of GrIS mass balance since 1840 using reanalysis products[92] show that recent trends are exceptional, with current rates of mass loss, accounting for inter-annual variability, higher than at any time in the last 180 years (Fig. 3). The post-1990 trend is in stark contrast to reconstructions of a positive mass balance[93] and ice sheet thickening[77] in the 1970s/early 1980s, suggesting that current losses are not likely to be part of a longer-term response/re-equilibration to an earlier period of warming. Indeed, the 30-year running mean (usually sufficient to remove meteorological variability) shows that, since the mid-1990s, this trend has become increasingly negative, reaching a minimum for the entire record in recent years. Moreover, there is strong consensus amongst experts that recent mass loss is largely (90%) driven by external forcing[94], pushing the ice sheet well beyond equilibrium[17] with a global mean

temperature of +1.2 °C above pre-industrial, and with early warning signs that parts of the GrIS may be close to a critical transition in terms of the melt-elevation feedback[50] (see Box 1). Hence, mass loss from the GrIS will continue, and is likely to accelerate, irrespective of any aspirational temperature targets that lie at or above +1.5 °C.

Reconstructions of ice sheet mass balance in Antarctica prior to 1992 are more challenging, due to its sheer size and because there are fewer proxies available compared to the GrIS[92]. The IPCC's First Assessment Report in 1990 highlighted mass loss in Greenland but noted that it was impossible to judge whether the Antarctic ice sheet was in balance or contributing negatively or positively to sea level[95]. Of the few studies available at that time, most argued that the ice sheet was in equilibrium or, like the GrIS had experienced modest mass gains in the 1970s and 1980s[96], leading the IPCC to conclude[95] that future warming should lead to increased accumulation and a negative contribution to GMSL. Most of the early altimetry data supported this assertion, revealing small increases in elevation in the early 1990s over the EAIS and, combined with long-term accumulation data, showed no sign of any imbalance over the 20th century[78]. Others argued that basal melting beneath ice shelves had been underestimated and that inclusion of more realistic melt rates would suggest it had been slowly losing mass in the 1980s[97]. This is consistent with a recent study[16] that extended the time series of mass balance back to 1979 and revealed modest mass loss from 1979 to 1990, but increasing thereafter and reaching −252 ± 26 Gt year⁻¹ for the period 2009–2017, largely driven by an acceleration in ice discharge from the Amundsen Sea Embayment (ASE) in West Antarctica since the 1990s. Notable mass loss from the ASE, which some have interpreted as a manifestation of marine ice sheet instability[52,53] (Box 1), has been attributed to grounding line retreat caused by the intrusion of warm, salty, circumpolar deep water (CDW) onto the continental shelf[14,16,51,53,79]. Importantly, there is emerging evidence of mass loss in East Antarctica, particularly in Wilkes Land[16,29,91,98], often referred to as the 'weak underbelly' of the EAIS[29,99]. This region contains almost 4 m of SLE and shares many characteristics with the WAIS, with deep subglacial basins and retrograde bed-slopes[9]. Mass loss has been detected for at least three decades[16,81] and has increased ten-fold[91] between 2003–2008 and 2016–2020, albeit from a low baseline. Similar to the WAIS, mass loss has been attributed to basal melting of ice shelves and grounding line retreat driven by intrusions of CDW[16,100–102], which has warmed by between 0.8 and 2 °C since the 1990s[103]. However, unlike the WAIS, these losses are currently offset by mass gains elsewhere in East Antarctica[98], which may be part of a long-term trend that has mitigated 20th Century SLR[104].

Despite a clear trend of overall mass loss from the WAIS (Fig. 2), experts were divided and uncertain about the relative contribution of internal variability and external forcing[94]. Recent work[105] suggests about 40% of total mass loss (2002–2021) can be attributed to El Niño/Southern Oscillation (ENSO) and persistent forcing from a more positive Southern Annular Mode (SAM) since the 1940s, likely driven by anthropogenic forcing over multi-decadal timescales. A more positive SAM reduces precipitation and therefore reduces surface mass balance, particularly over the EAIS[105]. A positive SAM and high ENSO also increase westerly winds, which increase the flow of CDW onto the continental shelf and contributes to ice shelf thinning[106,107]. Observations and climate models also suggest that increased greenhouse gas forcing has changed wind patterns that have enhanced the intrusion of CDW onto the Amundsen Sea shelf, consistent with increased mass loss from both Pine Island and Thwaites Glacier since the early 1990s[108]. The grounding lines of both glaciers were relatively stable throughout much of the Holocene[108], but retreat may have been initiated by a prolonged El Niño event in the 1940s[109,110], highlighting that short-term variations in ocean-climate conditions can lead to amplifying feedbacks that increase the sensitivity of ice sheets to climate change[83] and, in this case, trigger an episode of retreat that is thought irreversible on decadal timescales[111].

Other approaches to infer AIS mass balance further back in time have assumed that it can be treated as the residual in the sea level budget, indicating a negligible contribution (around +0.1 mm year⁻¹) over the 20th

century[4,5]. A number of proxy records[112] also suggest that both the rate of current SLR and its absolute magnitude are larger than they have been in the last 3000 years, supporting the inference of an anthropogenic signal driving at least part of the ice sheet contributions to GMSL.

## Numerical modelling: thresholds and irreversibility

Numerical ice sheet modelling is a powerful tool to explore the sea-level contribution from ice sheets under various forcing scenarios and the potential impact of crossing specific temperature thresholds[20–28,30,40,113]. Most modelling studies tend to focus on comparisons between the IPCC's low, medium and high emissions scenarios, where low is SSP1-2.6 (≈Representative Concentration Pathway (RCP) 2.6), corresponding to a total warming of around +1.8 °C above pre-industrial at 2100[1]. Fewer studies have explored the more aspirational target of the Paris Climate Agreement (+1.5 °C) and even fewer project ice sheet response to lower temperature targets and/or beyond the next century, when ice dynamical processes could emerge and contribute several metres to GMSL within a few centuries[1,27,28]. A recent Ice Sheet Model Intercomparison Project (ISMIP6)[25], for example, compared the future sea-level contribution of Greenland at 2100 under RCP8.5 (giving +9 ± 5 cm) and RCP2.6 (giving +3.2 ± 1.7 cm). Similarly, the ISMIP6 experiments[26] for Antarctica's sea-level contribution at 2100 focussed mainly on RCP8.5 (giving −7.6 to +30 cm of SLR) but with a small number of RCP2.6 experiments used to assess its response to more moderate forcing (giving −1.4 cm to +15.5 cm at 2100). A similar approach was taken in a more recent ISMIP6 study[114] that explored Antarctica's contribution to sea level up to 2300, recognising that instabilities are unlikely to emerge this century, but could develop and destabilise large parts of Antarctica over the next 300 years. Indeed, they found that while the sea-level contribution from Antarctica is relatively limited during the twenty-first century (less than +30 cm by 2100), it increases rapidly thereafter and reaches up to +1.7 and +4.4 by 2200 and 2300, respectively, under high emissions scenarios (SSP5-8.5) (and averaging +6.9 m for simulations that include ice shelf collapse). Two experiments used low-emission scenarios (SSP1-2.6) and gave a range of positive and negative sea-level contributions at 2300 (+46 cm to −36 cm), with inter-model differences relatively small until 2100 but increasing thereafter. Indeed, a key conclusion from this study was that the choice of ice sheet model remains the leading source of uncertainty in multi-century projections of Antarctic ice sheet response. Another recent study[115] found Antarctica's contribution to sea-level at 2300 under SSP1-2.6 ranged from +50 cm to −20 cm, but the longer-term commitment is much larger and reaches up to +6.5 m over the next millennia, even after climate forcing is stabilised at levels projected to be reached during this century under SSP1-2.6.

Importantly, it has been noted[90] that, despite recent advances, few ice sheet models accurately reproduce the rapid mass loss from ice sheets over the last few decades, suggesting uncertainties may be larger than assumed, even for low emissions scenarios. Indeed, while models are effective in exploring parametric uncertainty, they are less well suited for capturing epistemic uncertainties[1,94]. This point is highlighted by an analysis of expert judgement[94], which found much higher uncertainties associated with ice sheet contributions to sea level, e.g. a GMSL rise >2 m by 2100 fell within the 90th percentile credible range for a high emissions scenario, which is over twice the upper value reported in the IPCC's AR5[85].

Of the few studies that project ice sheet response to global mean temperatures at +1.5 °C or below (including the IPCC's 'very low' scenario: SSP1-1.9), there is a clear consensus that such scenarios will not halt SLR from Greenland and Antarctica (Table 2); and that even current forcing (+1.2 °C) might be sufficient to trigger instabilities that generate high rates of SLR (e.g. >10 mm year⁻¹) that would challenge adaptation measures.

Recent work on the AIS[27], for example, shows that median SLR accelerates to >1.5 mm year⁻¹ at 2100 under a +1.5 °C scenario (Fig. 4), with a total contribution of +8 cm at 2100 and a likely range (17th–83rd percentile) of +6 to +10 cm. This total contribution increases to +52 cm at 2200 (+22 to +77 cm) and +1.03 m (+0.61 to +1.22 m) at 2300, with no indication of a slow-down in the rate of SLR. Under a +2 °C warming

scenario, the rate of SLR increases to 2 mm year⁻¹ at 2100 and, at +3 °C of warming, this jumps to almost 5 mm year⁻¹, with RCP8.5 generating >15 mm year⁻¹ at 2100[27]. For comparison, another recent study[116] found rates of SLR from Antarctica that peak at around 4 mm year⁻¹ at around 2300, under conditions slightly warmer than present (RCP2.6), declining thereafter but maintaining a positive contribution to sea level until at least the year 3000, when the simulations ended. That study[116] also generated similarly high rates of SLR from Antarctica (15–20 mm year⁻¹ by 2300) using RCP8.5.

Numerical ice sheet modelling forced by the IPCC's most optimistic scenario (SSP1-1.9), where temperatures briefly exceed +1.5 °C and then stabilise around +1.4 °C by 2100, also show it would be insufficient to halt SLR from ice sheets (Table 2). Projections using a coupled atmospheric-ocean-ice-sheet model[30] found a total ice-sheet contribution of +20 ± 1 cm by 2150, despite stabilising feedbacks that included freshwater-induced atmospheric cooling around Antarctica. Their longer-term projections showed that only SSP1-1.9 avoids an acceleration in SLR towards 2500, but the sea-level contribution from both ice sheets continues to increase slowly[30]. Another study[117] found small but positive median sea-level contributions from both Greenland (+2 cm) and Antarctica (+4 cm) under SSP1-1.9 at 2100, but noted that pessimistic, yet plausible, projections incorporating processes leading to higher sea-level contributions under this SSP would see Antarctica's contribution increase fivefold by 2100 (Table 2). Over longer timescales, the committed median SLR from all sources (not just ice sheets) has been estimated at +0.7 m to +1.2 m, even if net zero greenhouse gas emissions are sustained until 2300, but ~3 m from Antarctica could not be ruled out[22]. Furthermore, each 5-year delay in near-term peaking of $CO_2$ emissions increased median SLR at 2300 by around +20 cm, and no net zero scenario gave a median SLR below +1.2 m at 2300 once global mean temperatures exceed +1.5 °C[22].

Even projections that maintain atmospheric and ocean climate forcing from 2020 that implement MICI (Box 1) but with no additional warming, can generate +1.34 m by 2500 (Fig. 4)[27], implying that a temperature forcing lower than present is required to halt SLR from ice sheets. Longer-term projections to the year 3000 have also used present-day forcing[116] and show a sea-level contribution from Antarctica that is around +1 m at that time. In this case, the rate of SLR peaks at around 2 mm year⁻¹ around 2400 and then declines, but it remains above zero until the end of the simulations. This is consistent with previous modelling of the AIS, which found that the WAIS does not regrow to its present configuration unless temperatures are cooled to 1 °C below pre-industrial[24]. Recent sensitivity experiments[118] also show that ocean-driven melt rates would need to be reduced to below present-day rates to promote re-advance of grounding lines in the ASE over the next two centuries, and that substantially increased accumulation would also be necessary to reverse its future sea-level contribution.

Some modelling studies, including those using ocean forcing from the 1990s[52], suggest that MISI (Box 1) may already be underway in the ASE[119], although the extent to which it is reversible is debated. A recent stability analysis indicated that present-day grounding line retreat is not yet irreversible[120], but current climate forcing, if sustained, leads to grounding line retreat that eventually becomes irreversible (referred to as 'committed tipping'), and leading to +2.7 to +3.5 of SLR over millennial timescales[121]. This is consistent with earlier work showing that 'present' ocean-climate forcing (in this case representing climate conditions in 2008)[122] is sufficient to drive a slow but sustained sea-level contribution from the ASE. Indeed, there are now multiple studies[36,116,121,123,124] indicating that recent ocean warming is likely to be sufficient to trigger millennial-scale collapse of parts of the WAIS, which would raise GMSL by several metres. Furthermore, submarine melting in newly formed ocean cavities, at rates similar to those in the 2000s, leads to a self-reinforcing feedback, which further accelerates grounding line retreat, even without ice-shelf collapse or MICI[122]. Even when melting is reduced to zero, some catchments continue to lose mass, implying that a tipping point may already have been passed[125]. Given recent projections of rapid ocean warming of the ASE this century at triple the historical rate[51], it seems clear that the initiation of at least partial collapse of

**Table 2 | Global mean sea-level contributions from Antarctica and Greenland under emissions/warming scenarios at + 1.5 °C or below**

| Reference | Scenario/ Temperature | SLR at 2100 (cm) Median (17th–83rd percentiles) | SLR at 2200 (cm) Median (17th–83rd percentiles) | SLR at 2300 (cm) Median (17th–83rd percentiles) | SLR at 2500 (cm) Median (17th–83rd percentiles) | SLR at 3000 Median (17th–83rd percentiles) |
|---|---|---|---|---|---|---|
| **Antarctic Ice Sheet** | | | | | | |
| Coulon et al. (2024)[116] | Present-day (2015 climatology) | 2 (−6 to 9)[a] | | 20 (−17 to 121)[a] | | 132 (−40 to 210) [a] |
| DeConto et al. (2021)[27] | 2020 forcing (+1.1 °C) | 5 | 20 | 75 | 134 | |
| Edwards et al. (2021)[117] | SSP1-1.9[b] | 4 (−1 to 10) | | | | |
| Park et al. (2023)[30] | SSP1-1.9 | 3 ± 0.8 | | | | |
| Edwards et al. (2021)[117] | SSP1-1.9 (risk averse) | 21 (12–32) | | | | |
| DeConto et al. (2021)[27] | +1.5 °C | 8 (6–10) | 52 (22–77) | 103 (61–122) | | |
| Fox-Kemper et al. (2021)[1] IPCC | SSP1-1.9 | 10 (3–25) | | | | |
| Fox-Kemper et al. (2021)[1] IPCC | SSP1-1.9 (low confidence) | 19 (2–56) | | | | |
| **Greenland Ice Sheet** | | | | | | |
| Edwards et al. (2021)[117] | SSP1-1.9 | 2 (−6 to 11) | | | | |
| Park et al. (2023)[30] | SSP1-1.9 | 12 ± 1 | | | | |
| Fox-Kemper et al. (2021)[1] IPCC | SSP1-1.9 | 5 (0–9) | | | | |
| Fox-Kemper et al. (2021)[1] IPCC | SSP1-1.9 (low confidence) | 18 (9–59) | | | | |

[a]Values in brackets refer to 5th–95th percentiles.
[b]SSP1-1.9 is the IPPC's 'very low' (lowest) scenario with a median temperature projection of +1.4 °C at 2100 (after a brief +1.4 °C overshoot) and with $CO_2$ concentrations at 440 ppm in 2100.

the WAIS is almost inevitable this century, even if it has not already been triggered.

GIA effects might provide a stabilising effect, slowing the pace of ice loss, whereby unloading of the lithosphere, causes bedrock uplift due to the viscoelastic response of the underlying mantle. In Greenland this could reduce the area of the ice sheet exposed to surface melting and marginal areas in direct contact with warm ocean water. However, due to the high mantle viscosity, uplift is relatively slow (mm to cm year$^{-1}$), with sensitivity analyses showing it has a negligible effect on centennial timescales and only reduces mass loss by ~2% on millennial timescales[23]. In Antarctica, GIA effects are more complex due to the enhanced vulnerability of grounding lines and the spatial heterogeneity of Earth structure, mantle viscosity, and uplift rates across the continent[126]. In the ASE, where current mass loss is concentrated[16,18,98], the thin lithosphere, very low-viscosity mantle, and uplift rates on the order of tens of mm year$^{-1}$ could provide a stabilising effect[127]. Indeed, recent modelling[128] combining a continental ice sheet model with a high-resolution GIA model that includes 3D lateral variations in Earth structure demonstrates that GIA feedbacks can slow WAIS loss under future warming, but the effect is inconsequential over the 21st century, as found in earlier studies using simpler models without viscous deformation[129] or 3D Earth structure[130]. On longer multi-century timescales, the GIA feedback becomes more important in slowing (by up to 40%) ice loss under RCP2.6[128]. However, under higher emissions scenarios, grounding-line retreat outpaces bedrock uplift and the GIA feedback becomes less effective as a natural 'brake', with SLR amplified by water expulsion from marine-based sectors[128]. Importantly, the thicker lithosphere and higher viscosity under the EAIS, limits the potential for uplift to slow ice loss in marine-based sectors[128,131] vulnerable to MISI[9,27,29,99].

Finally, a recent study[132] introduced a concept similar to the transient climate response in GCMs, termed the 'transient sea level sensitivity', defined as the multi-decadal to century response of GMSL to a unit change in temperature. An advantage of this approach is that it is agnostic to the emissions scenario and only depends on temperature relative to pre-industrial. Examining the historical record and CMIP6 simulations for the next century, the transient sea level sensitivity for the medium term is +5.3 ± 1 mm year$^{-1}$ °C$^{-1}$, i.e. +53 cm of SLR per century for a +1 °C of warming above pre-industrial or +80 cm for +1.5 °C of warming[132]. Crucially, the 'balance temperature' at which SLR equals zero lies close to pre-industrial temperatures[132]. This analysis was based on past ice sheet behaviour and numerical model simulations where the ice sheets do not experience instabilities (Box 1), but such instabilities become increasingly likely as temperatures continue to warm above pre-industrial.

## Summary and future perspectives: identifying a 'safe' temperature limit for ice sheets

The Greenland and Antarctic ice sheets store ~65 m of GMSL equivalent and even small changes in their volume will profoundly alter coastlines around the world, displacing hundreds of millions of people and causing loss and damage well beyond the limits of adaptation. This existential threat has been known since the 1970s, prompting research to ascertain the temperature thresholds that might push them beyond equilibrium and into a predominantly negative mass balance that increases GMSL. Of concern is that the best estimates of these thresholds have lowered over recent decades. Early modelling of the EAIS, for example, suggested it might take +17–20 °C of warming to destabilise the ice sheet[133], a view echoed in the IPCC's AR3[134]; whereas recent estimates suggest marine-based sectors might be vulnerable to +2 to +3 °C of warming[29,44]. Likewise, best estimates for the GrIS have lowered from +3.1 °C[135] to +1.6 °C[136] and, most recently, to +1.5 °C, which is also viewed as a 'tipping point' for the WAIS[44]. This improved knowledge of ice sheet sensitivity to climate change has motivated calls to meet the most ambitious target in the Paris Climate agreement and limit warming to +1.5 °C.

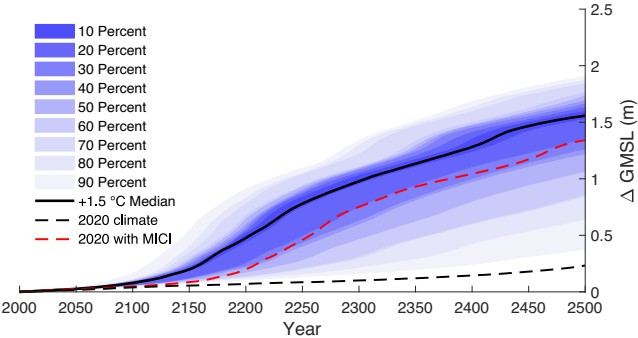

**Fig. 4 | Antarctica's contribution to future sea level with+ 1.5 °C scenario versus a scenario with no warming after 2020.** The blue shading shows the time-evolving uncertainty (percentiles) around the median (black line) from an observationally calibrated ice sheet model[27] ensemble using a range of model parameters and +1.5 °C warming scenario. The red dashed line is from the same ice sheet model[27] with model parameters representing the median values used in the +1.5 °C ensemble and including MICI (see Box 1), but with no additional warming beyond 2020. The black-dashed line uses the same model[27] and 2020 climate scenario shown in red, but without MICI.

It is widely acknowledged that these policy-relevant temperature targets, such as those adopted in the Paris Agreement, are not necessarily inherent and precise thresholds that cannot be exceeded without catastrophe[137,138]. Indeed, some have argued that the discourse around 'tipping points' has the potential to confuse and distract from urgent climate action, in part because there remain large uncertainties about where various tipping points might lie[138]. Despite these concerns, however, such policy targets serve as valuable benchmarks to limit cumulative harm[138], whilst also recognising that every fraction of a degree of warming will have important consequences.

With respect to ice sheets, a warming target of +1.5 °C (even with a temporary over-shoot) is an admirable goal, with a best-case scenario that it will result in a steady increase in the rate of GMSL, but with no evidence to suggest it will halt or even slow the rate of SLR from the world's ice sheets. Today's climate (+1.2 °C above pre-industrial) is already generating substantial mass loss, which has quadrupled since the 1990s[1,18], and with multiple studies[1,17,22,27,46,139–141] indicating that policymakers should now prepare for several metres of SLR over the coming centuries. Furthermore, there is a growing body of evidence that current climate forcing could trigger rapid retreat in both Greenland[50] and West Antarctica[121,122], which some argue has already been initiated[52,53,125]. Given that ice sheet 'tipping points' will probably not be apparent until after they have been passed[142], it is imperative to adopt the precautionary principle. To that end, one might ask: what is a 'safe' limit for ice sheets?

Answering this question is very challenging, not least because so few studies have projected the response of ice sheets to lower-than-present temperature forcing[143], but it is worth exploring in light of the preceding review. Evidence from the palaeo-record clearly shows that a global mean temperature that exceeds +1 °C above pre-industrial leads to several metres of sea level rise (Table 1), with higher absolute values of SLR becoming increasingly likely the higher the warming, and the longer it is sustained[76]. Observations and reconstructions of ice sheet mass balance also appear to indicate a clear trend of increasing mass loss over the last few decades that is unprecedented in at least the last 3,000 years[112]. Therefore, the combined evidence from the palaeo-record, recent observations of ice sheet mass balance and numerical modelling, all indicate the need to return to cooler-than-present conditions to slow SLR from ice sheets and prevent a rapid acceleration (e.g. to rates >10 mm year$^{-1}$) that would stretch the limits of adaptation. As noted, determining a temperature target to avoid a rapid increase in SLR from ice sheets is challenging and further work is urgently required to explore the impacts of strong mitigation and lower global warming levels, particularly to inform vulnerable coastal and island states[143].

However, we hypothesise that it probably lies at, or below, +1.0 °C above pre-industrial, which is similar to conditions in the 1980s when ice sheets were broadly in balance. As noted elsewhere[144], this warming level is consistent with a safe limit proposed in the early 1990s[145] and a planetary 'boundary' of $CO_2$ concentrations that should not exceed 350 ppm[146], reinforcing recent calls for a stricter and more ambitious temperature target for a safe and just future for planet Earth[48,144,147].

## Data availability

The GRACE data used in Figs. 2 and 3 are publicly available from https://data1.geo.tu-dresden.de/gis_gmb/ (ref. 148). All other data used in this manuscript is clearly attributed to the source publication via citation. No new datasets were generated during the study.

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

## Acknowledgements

C.R.S. acknowledges funding from the Natural Environment Research Council (NE/R000824/1). We are grateful for the constructive comments from three anonymous reviewers on an earlier version of this manuscript.

## Author contributions

C.R.S. developed the idea for the paper and J.L.B. provided input on its initial contents and structure. C.R.S. drafted the first section, with input and edits

from all authors. A.D. drafted the second section with input and edits from all authors. J.L.B. drafted the third section with input and edits from all authors. C.R.S. drafted the fourth and fifth sections with input and edits from all authors. A.D. undertook the literature review to produce Table 1 and Fig. 1; J.L.B. produced Figs. 2 and 3 and R.M.DeC. contributed data to produce Fig. 4. All authors provided final comments and edits on all sections of the paper.

## Competing interests

The authors declare no competing interests.
