## [Transparent Peer Review file · Communications Earth & Environment]

Warming of +1.5 °C is too high for polar ice sheets

Corresponding Author: Professor Chris R Stokes

This manuscript has been previously reviewed at another journal. This document only contains information relating to versions considered at Communications Earth & Environment.

Version 0:

Decision Letter:

Dear Professor Stokes,

Your revised manuscript titled "Why +1.5 °C is too warm for polar ice sheets" has now been seen by the original reviewers 2 and 3, whose comments appear below. In light of their advice we are delighted to say that we are happy, in principle, to publish a suitably revised version in Communications Earth & Environment.

We therefore invite you to revise your paper one last time to address the remaining concerns of our reviewers. At the same time we ask that you edit your manuscript to comply with our format requirements and to maximise the accessibility and therefore the impact of your work.

EDITORIAL REQUESTS:

****Please take care to match our formatting and policy requirements. We will check revised manuscript and return manuscripts that do not comply. Such requests will lead to delays. ****

SUBMISSION INFORMATION:

OPEN ACCESS:

Communications Earth & Environment is a fully open access journal. Articles are made freely accessible on publication. For further information about article processing charges, open access funding, and advice and support from Nature Research, please visit <https://www.nature.com/commsenv/open-access>

Link Redacted

Best regards,

Alireza Bahadori, PhD
Associate Editor
Communications Earth & Environment
Consulting Editor
Communications Sustainability

REVIEWERS' COMMENTS:

Reviewer #2 (Remarks to the Author):

I appreciate the authors' thorough revisions, which have significantly improved the manuscript while preserving its core message. I also commend their restraint in avoiding overstatements and unnecessary dramatization. Clearly, the previous title and abstract did not serve the rest of the paper well, and I am pleased to see these aspects improved. Additionally, I am grateful for the use of SL rates instead of SL, as this allows for a more precise quantification of impacts. Overall, the paper reads very well, and I am happy with the outcome. I have only a few minor comments where additional context could be beneficial.

Lines 79–82: Regarding the statement that rapid SLR rates from self-reinforcing feedback may be triggered by even a small temperature increase and that follows the discussion on the rapid deglaciation of MWP1A: It is important to note that MWP1A was an SLR event involving a larger number of ice sheets than those currently present or those that existed during previous interglacials. Naturally, if more ice sheets lose mass, the rate of SLR will be higher for a given temperature rise. Therefore, one should be careful with a direct comparison of potential impacts.

The same is valid for lines 119-122: The Laurentide ice sheet was also significantly bigger than Greenland, therefore a direct comparison with rates of SL during deglaciation of the Laurentide to the Greenland ice sheet should be done with care.

Line 399: I would prefer to change 'dramatic mass loss' in either significant or important mass loss.

Reviewer #3 (Remarks to the Author):

I previously reviewed an earlier version of this manuscript and raised concerns, particularly regarding the headline statements on temperature thresholds and irreversibility. I am pleased to see that the authors have addressed these issues, and I am now very happy with the paper's framing. It provides an excellent review of the topic and presents a well-balanced perspective on the current state of research. I have only a few very minor comments that I hope the authors will consider.

L460-461: I think it's worth adding a brief health warning at the end of this statement emphasizing that such high projections rely almost entirely on the model implementation of MICI, which has been questioned in other recent studies.

L566-568: Despite the hint of a caveat at the start of the sentence, I would still maintain that this is not a credible line of evidence as mentioned in my previous review and is at odds with many other statements in the paper regarding ice sheet response timescales.

I congratulate the authors on an excellent study.

Response to Review Comments

Reviewer #2 (Comments for the Author):

I appreciate the authors' thorough revisions, which have significantly improved the manuscript while preserving its core message. I also commend their restraint in avoiding overstatements and unnecessary dramatization. Clearly, the previous title and abstract did not serve the rest of the paper well, and I am pleased to see these aspects improved. Additionally, I am grateful for the use of SL rates instead of SL, as this allows for a more precise quantification of impacts. Overall, the paper reads very well, and I am happy with the outcome. I have only a few minor comments where additional context could be beneficial. We thank the reviewer for taking the time to review our revised manuscript and are pleased that they are happy with the outcome.

Lines 79–82: Regarding the statement that rapid SLR rates from self-reinforcing feedback may be triggered by even a small temperature increase and that follows the discussion on the rapid deglaciation of MWP1A: It is important to note that MWP1A was an SLR event involving a larger number of ice sheets than those currently present or those that existed during previous interglacials. Naturally, if more ice sheets lose mass, the rate of SLR will be higher for a given temperature rise. Therefore, one should be careful with a direct comparison of potential impacts. Amended: we have edited the sentence to acknowledge that there were a larger number of palaeo-ice sheets than at present.

The same is valid for lines 119-122: The Laurentide ice sheet was also significantly bigger than Greenland, therefore a direct comparison with rates of SL during deglaciation of the Laurentide to the Greenland ice sheet should be done with care. The key point here is that the ice-elevation feedback mechanism ('saddle collapse') has been invoked to have caused rapid sea-level rise. We are not making a direct comparison with SLR from the Greenland Ice Sheet, but simply that the same positive mechanism is being invoked in central West Greenland.

Line 399: I would prefer to change 'dramatic mass loss' in either significant or important mass loss. Amended to 'rapid mass loss'.

Reviewer #3 (Comments for the Author):

I previously reviewed an earlier version of this manuscript and raised concerns, particularly regarding the headline statements on temperature thresholds and irreversibility. I am pleased to see that the authors have addressed these issues, and I am now very happy with the paper's framing. It provides an excellent review of the topic and presents a well-balanced perspective on the current state of research. I have only a few very minor comments that I hope the authors will consider. We thank the reviewer for taking the time to review our revised manuscript and are pleased that they now find it acceptable.

L460-461: I think it's worth adding a brief health warning at the end of this statement emphasising that such high projections rely almost entirely on the model implementation of MICI, which has been questioned in other recent studies. Amended: we have edited the sentence to make it clear that MICI is implemented and have cross-referenced to Box 1, where there is already a clear acknowledgement of the debate, i.e. "The validity and implementation of MICI is debated^{1,59} and, unlike MISI, there is no evidence that it is currently underway, with one recent study showing it might not occur this century⁵⁹."

L566-568: Despite the hint of a caveat at the start of the sentence, I would still maintain that this is not a credible line of evidence as mentioned in my previous review and is at odds with many other statements in the paper regarding ice sheet response timescales. Amended: we have deleted this sentence. However, we simply wanted to make the point that we do not have to go too far back in time to a climate state that is likely to be much safer for ice sheets and so we have moved that point to the penultimate sentence of the paragraph. It is simply an observation and may be relevant but

we agree that it is not the strongest line of evidence.

I congratulate the authors on an excellent study. Thank you!